

# Plasma proteomic analysis of systemic lupus erythematosus patients using liquid chromatography/tandem mass spectrometry with label-free quantification

Rashmi Madda[1], Shih-Chang Lin[2,3], Wei-Hsin Sun[1] and Shir-Ly Huang[4]

[1] Department of Life Sciences, National Central University, Zhongli, Taiwan
[2] Division of Medicine, College of Medicine, Fu Jen Catholic University, Taipei, Taiwan
[3] Department of Rheumatology and Immunology, Cathay General Hospital, Taipei, Taiwan
[4] Institute of Microbiology and Immunology, National Yang-Ming University, Taipei, Taiwan

## ABSTRACT

**Context:** Systemic lupus erythematosus (SLE) is a chronic inflammatory autoimmune disease with unknown etiology.

**Objective:** Human plasma is comprised of over 10 orders of magnitude concentration of proteins and tissue leakages. The changes in the abundance of these proteins have played an important role in various human diseases. Therefore, the research objective of this study is to identify the significantly altered expression levels of plasma proteins from SLE patients compared with healthy controls using proteomic analysis. The plasma proteome profiles of both SLE patients and controls were compared.

**Methods:** A total of 19 active SLE patients and 12 healthy controls plasma samples were analyzed using high-resolution electrospray ionization liquid chromatography-tandem mass spectrometry (LC-ESI-MS/MS) followed by label-free quantification.

**Results:** A total of 19 proteins showed a significant level of expression in the comparative LC-ESI-MS/MS triplicate analysis; among these, 14 proteins had >1.5- to three-fold up-regulation and five had <0.2- to 0.6-fold down-regulation. Gene ontology and DAVID (Database Annotation Visualization, and Integrated Discovery) functional enrichment analysis revealed that these proteins are involved in several important biological processes including acute phase inflammatory responses, complement activation, hemostasis, and immune system regulation.

**Conclusion:** Our study identified a group of differentially expressed proteins in the plasma of SLE patients that are involved in the imbalance of the immune system and inflammatory responses. Therefore, these findings may have the potential to be used as prognostic/diagnostic markers for SLE disease assessment or disease monitoring.

Corresponding authors
Wei-Hsin Sun,
weihsin@cc.ncu.edu.tw
Shir-Ly Huang, sl.huang@ym.edu.tw

## INTRODUCTION

Systemic lupus erythematosus (SLE) is a chronic autoimmune disease involving multiple organs that are characterized by excessive production of antinuclear antibodies (*Crispín et al., 2010*). The particular reasons behind the cause of SLE have not been recognized yet, but numerous factors have been shown to be related to the development of SLE, including genetic factors, environmental triggers, hormonal imbalance, and life cycle changes (*Lisnevskaia, Murphy & Isenberg, 2014*). One of the most important mechanisms involved in the development of SLE is alteration to immune reactions; this involves auto-antibodies that target the patient's own tissues and subsequently lead to inflammation (*Pisetsky, 1997*).

Rapid advances in proteomic technology have led to the identification of several novel protein biomarkers related to both SLE and lupus nephritis (LN); these include monocyte chemoattractant protein-1 (MCP-1) (*Marks et al., 2010*), the tumor necrosis factor-like weak inducer (TWEAK) (*Schwartz, Michaelson & Putterman, 2007*), transferrin (*Suzuki et al., 2007*), various interleukins and TNF-α (*Almoallim et al., 2012*) proteins. However, these organ-specific biomarkers do not serve only as markers of lupus and of primary organ participation in lupus. Despite tremendous basic and clinical research progress regarding biomarkers discovery, SLE remains an unsolved puzzle due to a lack of appropriate disease monitoring and the absence of predictive/diagnostic biomarkers. Thus the quest for such markers is still continuing.

Human serum/plasma holds abundant information on the physiological and pathological states of an individual's body, and as a result, such samples can provide valuable clinical parameters (*Anderson & Anderson, 2002*). Consequently, they have enormous potential in terms of revealing disease conditions and they also hold the promise of a revolution in disease diagnosis and therapeutic monitoring. Moreover, plasma is a very rich source of proteins and tissue leakage with the presence of interleukins (*Anderson & Anderson, 2002*). The protein alterations in a diseased plasma sample may provide clues and potential clinical parameters that should help us to understand both the pathogenesis of a disease, as well as diagnosis.

Significant advances in proteomic technologies now allow the comprehensive profiling of protein expression levels in biofluids from patients with a given disease (*Hu, Loo & Wong, 2006*). Furthermore, mass spectrometry technologies have become a significant approach in clinical proteomics which allows the exploration in depth of an illness and its underlying mechanisms. Tandem mass spectrometry (MS/MS) coupled with multidimensional liquid chromatography (LC) together with database searching has emerged as a robust technique that allows protein identification and characterization. Numerous studies have successfully demonstrated the utility of high-resolution electrospray ionization liquid chromatography-tandem mass spectrometry (LC-ESI-MS/MS), which allows the highly sensitive identification of hundreds of distinct proteins in a given biomedical sample (*Korte, Gaffney & Powell, 2012*).

Therefore, our current study incorporated LC-ESI-MS/MS label-free shotgun proteomic methodology with the aim of identifying significant changes in the expression

**Table 1 General and demographic characteristics of the collected SLE patients and healthy controls plasma samples.**

| | SLE patients | Healthy individuals |
|---|---|---|
| Number of samples | 19 | 12 |
| Female: male (% female) | 18:1 (94.7%) | 9:3 (75%) |
| Age (years) | 32.1±1.5[a] | 32.9±2.1[a] |
| SLEDAI score (average) | 8.47 ± 2.8[a] | N/A |
| Anti-ds DNA antibodies | | |
| <30 ($n = 19$) | 2 (10.52%) | N/A |
| 30 to <60 ($n = 19$) | 2 (10.52%) | N/A |
| 60–200 ($n = 19$) | 7 (36.8%) | N/A |
| >200 ($n = 19$) | 8 (42.1%) | N/A |
| Anti-nuclear antibodies (ANA) | ≥1:640 | N/A |

**Notes:**
N/A, not applicable.
[a] Data are represented as mean ± standard deviation.

levels of plasma proteins from SLE patients compared with healthy controls. The proteins identified as having altered expression levels may serve as potential candidate biomarkers for SLE disease assessment and monitoring of the disease progression.

## MATERIALS AND METHODS

### Collection of SLE samples and healthy control samples

A total of 19 SLE patients (female/male; 18/one; mean age 32.1 ranging from 22 to 54 years) and the same age and gender-matched 12 healthy controls (female/male; nine/three; mean age 32.9 ranging from 26 to 54 years of age) gave plasma samples that were obtained from the Cathay General Hospital (CGH), Taipei, Taiwan. The disease assessment of the collected SLE samples was performed according to the classification criteria of the American College of Rheumatology (*Tan et al., 1982*). The Systemic Lupus Erythematosus Disease Activity Index (SLEDAI) scoring method was used to determine the level of SLE disease activity (*Bombardier et al., 1992*). Based on the disease activity index parameter results, all of the recruited SLE patients had active disease (SLEDAI more than eight) with high titers of anti-ds DNA antibodies, and the presence of proteinuria and low levels of complement. The demographic characteristics of patients and controls are presented in Table 1. Plasma was prepared from the whole blood samples, which were collected in heparin-containing tubes from both the SLE patients and the healthy controls. All of the patient samples were collected prior to the clinical treatment. The protein concentrations of the collected samples were determined by Bradford protein assay (*Zor & Selinger, 1996*) and the samples were stored at −80 °C until further analysis. This study obtained approval from the Institutional Review Board (IRB) (Approval code CT-09905) for Research Ethics at the Cathay General Hospital (CGH), Taipei, Taiwan. Informed consent was provided by all blood donors.

## Depletion of high abundance serum albumin protein

In order to identify the various low and high abundance proteins by mass-based proteomic analysis, highly abundant serum albumin was depleted from the SLE and control samples by using a dye-based proteo-prep blue albumin removal kit (Thermo Fisher Scientific, Waltham, MA, USA). Using the manufacturer's protocol, spin columns were suspended with slurry and centrifuged at $8{,}000 \times g$ for 10 s. Next, 0.1 ml of serum samples were added to the spin columns and the columns were incubated for 10 min. Subsequently, the samples were centrifuged at $8{,}000 \times g$ for 60 s; this was repeated twice to remove further serum albumin (*Ahmed et al., 2003*). The protein concentrations of the depleted serum samples were determined by Bradford assay and the samples were stored at $-80\ ^\circ$C until further analysis.

## Protein precipitation and in-solution digestion

The depleted SLE and healthy plasma samples were precipitated using 100% ice-cold acetone and then kept overnight at $-20\ ^\circ$C. Then the samples were centrifuged at $14{,}000 \times g$ for 10 min, and the pellets were dissolved in 100 μl of 25 mM $NH_4HCO_3$ and 6.5 M urea (0.1–1 μg/μl); this was followed by an in-solution digestion procedure used in previous studies (*Ru et al., 2006*). Next, the protein samples were reduced with 100 mM DTT at $37\ ^\circ$C for 30–40 min and alkylated with 200 mM IAA in the dark at room temperature for 25–35 min. After this, the protein samples were digested with sequencing grade trypsin (Promega, Madison, WI, USA) using a 50:1 ratio at $37\ ^\circ$C. The reaction was quenched by adding 2 μl of 50% formic acid (FA), the mixture vortexed and then centrifuged in order to collect the peptides. Finally, the solution was lyophilized and desalted using a C18 zip-tip procedure.

## Nano UPLC and mass spectrometry conditions

ESI interface Q-TOF MS/MS was carried out at a resolution of about 10,000 full-width half maximum performance. In order to calibrate the instrument, an external standard of lock mass BSA was continuously infused at a constant flow rate of 0.25 μl/min using the Nano-ACQUITY auxiliary pump at an interval of 20 s (lock spray frequency). The precursor mass error was less than 2 ppm and for accuracy, the lock mass data were averaged. The acquired peptide spectra were eluted using the positive V mode with a scan mass range of 50–2,000 *m/z* and a scan time of 1 s. After reconstituting in 3% ACN and 0.1% FA, the digested 400 ng peptides were injected into an online nano-ACQUITY, UPLC coupled Q-TOF, Synapt-HDMS mass spectrometer (Waters Corporation, Milford, MA, USA). The peptides were separated using a $C_{18}$ reverse phase column (1.7 μm × 75 μm × 250 mm) (Waters Corporation, Milford, MA, USA). The binary solvent system used consisted of 99.9% water and 0.1% FA (mobile phase A) and 99.9% ACN and 0.1% FA (mobile phase B). The peptides were initially pre-concentrated and desalted online at a flow rate of 5 μl/min using a 5 μm symmetry $C_{18}$ trapping column (internal diameter 180 mm, length 20 mm) (Waters Corporation, Milford, MA, USA) with 0.1% FA. After injection, the peptides were eluted into the Nano-LockSpray ion source at a flow rate of 300 n/l and a gradient of 2–40% for 120 min. Then the column was washed and

equilibrated. The digested plasma samples were run in triplicate and the data were analyzed by ProteinLynx Global Server 4.2 software (PLGS; Waters Corporation, Milford, MA, USA) (*Aggarwal et al., 2017*). Each sample was injected three times to obtain technical triplicates.

## Label-free quantification

To quantify the proteins from the LC-MS/MS, a label-free quantification analysis was performed using PEAKS Studio 8.0 (Bioinformatics Solutions Inc., Waterloo, ON, USA) (*Zhang et al., 2012*). Independent samples from each triplicate analysis were studied and compared between patients and the controls. Total raw data files were imported and processed using the Peaks software program for the interpretation of spectra and the retention time was set from 600 to 10,500 s. An in-house constructed Uniprot's reference database of *Homo sapiens* (release 03_2014) contained 20,272 entries was added and combined with a decoy database (the sequences were reversed). For label-free quantification the following parameters were specified: enzymatic digestion by trypsin, with two missed cleavages; precursor mass tolerance was 10 ppm; fragment mass tolerance: 0.7 Da, minimum charge: 2, maximum charge: 3. The specified fixed and variable modification consisted of carbamidomethylation (Cys), oxidation (M), and deamidated (N and Q). To determine the false-positive identification rate, the estimated spectra was used against decoy database. A false discovery rate (FDR) of $\leq 1\%$, with a peptide score of $-10 \log p \geq 20$ was considered adequate for confident protein identification. To determine the relative protein and peptide abundance in the tested samples, peptide feature based quantification was performed. The signal intensity of a peptide is directly proportional to the abundance of the peptide in the sample, therefore, the confidently identified peptide features were matched and the peptide intensity differences between two samples were able to be estimated. Likewise, the area under the curve of the extracted ion chromatograms (XICs) was measured and compared between two analyzed runs. To get the summed cumulative peak area of the protein, only unique peptides that are assigned to particular proteins were selected.

The FDR was calculated based on the target/decoy database, and the peptides with an FDR of $\leq 1\%$ were chosen as true positive hits (considering the risk of having one false positive in 20 observations). By using this active feature based quantitative approach the detected peptides with $p$-values $<0.05$ and 0.01 which were identified in at least three observations from the SLE samples compared to control samples were considered. In order to identify the significant protein differential expressions an independent sample $T$-test was performed. The quantified datasets were normalized using their spectral abundance factor values (the average of the triplicate experiments) and this was used to generate a heat map showing the differentially expressed proteins between the two groups.

## Quantification of altered proteins by emPAI

To acquire confident results from our proteomic study, the widely used exponentially modified protein abundance index (emPAI) was employed as an alternative method in

order to quantify the differences in abundances of the identified proteins between the SLE patients and the control subjects. Numerous label-free relative quantification studies have successfully applied the formula derived for emPAI and developed by *Ishihama et al. (2005)*. Specifically, the MASCOT database results were generated in an Excel file (composed of emPAI, protein description, peptide identifications, the charge states of the peptides, etc.), and this was uploaded to the freely accessible (http://empai.iab.keio.ac.jp) web-based system (http://empai.iab.keio.ac.jp) for emPAI evaluation. The following parameters were specified on the emPAI website: IPI_HUMAN database, trypsin digestion as the enzyme, carbamidomethyl (Cys) fixed modifications, mass range 500–3,000 Da, and no retention time filtering. The uploaded sample files were analyzed and the results were exported to a separate file. To estimate the protein abundances, the mean emPAI values of the triplicate MS analyses from patients and controls were calculated and Hochberg and Benjamini calculation of FDR ($p < 0.05$) was applied to correct for multiple testing errors. Protein level fold-difference was calculated from the relative means of normalized emPAI values and the fold change differences were estimated from all replicates, with the Student's *t*-test (two-tailed, heteroscedastic) applied to the same values. A minimum of three unique peptide sequences in at least three replicates were required for quantification.

## Protein identification

For the identification of proteins, the UniProtKB database (UniProt release 2015-10) and National Center for Biotechnology non-redundant (NCBInr) was used for the database search using the Mascot software (Matrix Science version 2.2, http://www.matrixscience.com) search engine. The following parameters were specified for searching altered proteins: enzymatic digestion by trypsin with two missed cleavages, followed by carbamidomethyl as fixed and oxidation (M) as variable modifications. The peptide mass tolerance was set at 50 ppm and the MS/MS tolerance was set at 0.1 Da with an FDR of $\leq 1\%$. Proteins identified in all three technical replicates or in at least two of the three analyses were considered to be identified and their theoretical molecular mass (MW) and isoelectric point (pI) were determined using Mascot database.

## Bioinformatics analysis

To interpret the biological processes (BPs), molecular functions (MFs), and cellular components (CCs) of the identified proteins, the international standardized gene function classification system of gene ontology (GO) (http://www.geneontology.org/), and the DAVID (http://david.abcc.ncifcrf.gov/) (Database Annotation Visualization, and Integrated Discovery) database with for functional analysis were used (*Huang, Sherman & Lempicki, 2009*). For the protein–protein interaction (PPI) networks among these proteins, the STRING (Search Tool for the Retrieval of Interacting Genes/Proteins, Version 9.1) database at the website: http://string-db.org/ with default parameters were used.

## Statistical analysis

The LC-ESI-MS/MS data was measured using peptide feature based and spectral counting based quantification in order to identify the differences in protein expression levels. Data are expressed as the mean ± standard deviation and differences were determined using two independent sample $t$-test and a paired $t$-test were used to determine the differentially regulated proteins between SLE and healthy controls. Statistical analysis were performed using the SPSS statistical package (SPSS16; SPSS Ltd., Working, and Surrey, UK) for Windows. A probability value of less than 0.05 with 95% confidence limit was considered as statistically significant and of $p$-values <0.01 were considered as highly significant.

## RESULTS

Screening for novel protein biomarkers from human biological fluids is able to provide essential clinical information and such information needs to be revealed via a range of proteomic strategies. Here, we have attempted to use LC-ESI-MS/MS to identify proteins that show significant alterations in expression levels among 19 SLE patients compared to 12 healthy controls. In order to detect differentially expressed proteins in the individual SLE plasma samples compared to controls, the samples were analyzed in triplicates by LC-ESI-MS/MS with label-free quantitative analysis was performed using PEAKS Q 8.0 software. Apart from the use of this software, an emPAI method by Ishihama et al. was also used manually to obtain highly confident proteomic data. Both the quantitative analysis results were complimentary to each other in terms of protein abundance and the fold change. Table 2 shows the details about the number of MS/MS spectra, peptide and protein identification and quantitation of each SLE patient and healthy sample. The proteins which are identified in at least two of the three technical replicates were selected for the quantitation and statistical analysis.

A total of 122 proteins (homologs/same name proteins were eliminated) from the SLE patients were identified (Tables S1 and S7), and 143 proteins were identified from the healthy controls with the molecular masses ranging from 9 to 550 kDa and isoelectric point ranging from 3.4 to 9.0. Out of the 122 proteins, 19 were found to have significant changes in their expression levels, including 14 up-regulated and five down-regulated; all of these fulfilled the appropriate statistical criteria (either $p < 0.01$ or $p < 0.05$). A comparative LC-MS/MS base peak intensity (BPI) chromatograms of SLE and healthy controls were shown in Fig. 1. The BPI chromatograms show the number of marked variations in the patient group than in the control group. All of the proteins were identified by the presence of more than two and up to ten unique peptides with a protein score of >70, and a significance score of <20. The complete list of identified proteins and their fold change differences with both the quantitative analysis data among SLE patients and healthy controls were presented in Table S2. The PEAKS Q software generated a heat map of the differentially expressed proteins by comparing the patients and controls were shown in Fig. 2. The FDR specified for the identified proteins was <1%. The complete comparative proteomic analysis using PEAKS 8.0 and emPAI quantitative evaluations of the up and down-regulated proteins were presented in Table 3.
**Table 2 Complete LC-MS/MS proteomic analysis details for peptide and protein identification and quantitation in each patient and healthy sample.**

| SLE patients | Total spectra* | Distinct peptides* | FDR spectra | FDR distinct peptide (%) |
|---|---|---|---|---|
| P1 | 169,115 | 38,918 | 0.41 | 1.21 |
| P2 | 176,318 | 46,186 | 0.43 | 1.24 |
| P3 | 165,812 | 35,981 | 0.47 | 1.29 |
| P4 | 126,050 | 38,843 | 0.45 | 1.34 |
| P5 | 169,295 | 37,918 | 0.41 | 1.49 |
| P6 | 173,378 | 36,186 | 0.43 | 1.26 |
| P7 | 165,812 | 39,991 | 0.44 | 1.25 |
| P8 | 126,050 | 28,843 | 0.42 | 1.26 |
| P9 | 149,295 | 37,918 | 0.42 | 1.27 |
| P10 | 173,378 | 36,186 | 0.43 | 1.28 |
| P11 | 161,812 | 35,881 | 0.45 | 1.44 |
| P12 | 126,050 | 28,843 | 0.44 | 1.33 |
| P13 | 169,295 | 37,918 | 0.43 | 1.35 |
| P14 | 183,878 | 38,186 | 0.42 | 1.26 |
| P15 | 165,812 | 35,981 | 0.45 | 1.24 |
| P16 | 136,050 | 28,843 | 0.45 | 1.27 |
| P17 | 169,295 | 47,918 | 0.45 | 1.25 |
| P18 | 173,378 | 36,186 | 0.43 | 1.26 |
| P19 | 165,812 | 45,981 | 0.42 | 1.33 |
| **Overall** | **3,045,885** | **712,707** | **8.25** | **24.62** |
| **Healthy controls** | **Total spectra** | **Distict peptides** | **FDR spectra** | **FDR distinct peptide (%)** |
| H1 | 138,215 | 48,118 | 0.31 | 1.24 |
| H2 | 196,228 | 56,286 | 0.43 | 1.23 |
| H3 | 195,912 | 65,181 | 0.47 | 1.31 |
| H4 | 177,050 | 47,143 | 0.44 | 1.34 |
| H5 | 199,295 | 57,928 | 0.38 | 1.39 |
| H6 | 189,378 | 39,196 | 0.41 | 1.22 |
| H7 | 165,912 | 41,991 | 0.41 | 1.24 |
| H8 | 177,550 | 39,823 | 0.38 | 1.24 |
| H9 | 159,295 | 39,938 | 0.41 | 1.27 |
| H10 | 193,378 | 39,186 | 0.44 | 1.28 |
| H11 | 181,702 | 38,981 | 0.45 | 1.44 |
| H12 | 152,080 | 38,643 | 0.41 | 1.33 |
| Overall | 2,125,995 | 552,414 | 4.94 | 15.53 |

Note:
* Total spectra observed for protein groups.

## Protein profile of the differentially expressed proteins

Based on the bioinformatics analysis, the identified differentially expressed proteins were found to be immunoglobulins, acute phase reactants, glycoproteins, transporters, antiproteases, and binding proteins. Mounting evidence reveals that most of the identified

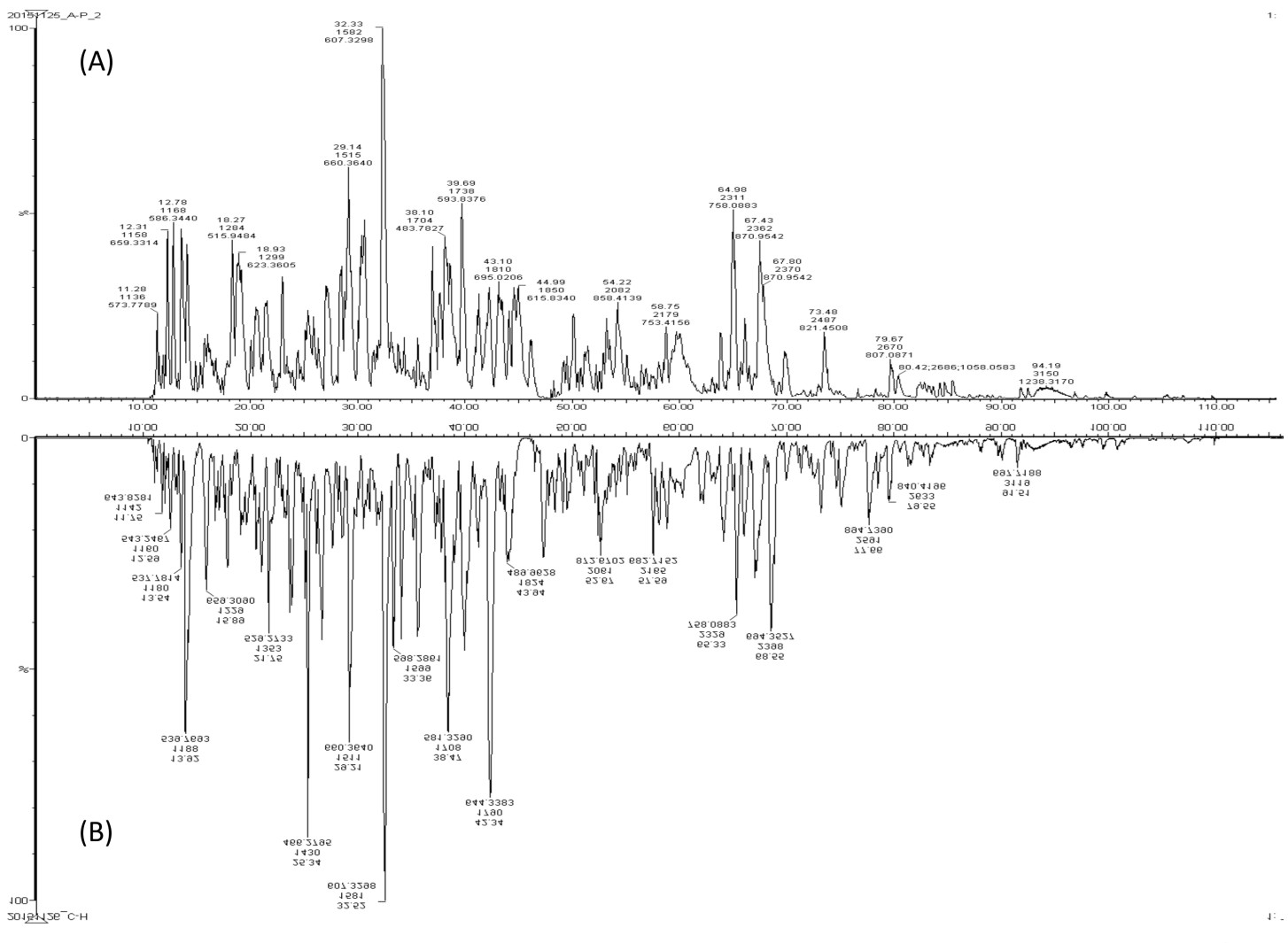

**Figure 1 A representative LC-MS/MS base peak chromatogram.** (A) SLE patients compared to (B) healthy controls.

proteins play crucial roles in immune system regulation, inflammation, and acute inflammatory responses. SLE patients were found to have a 2.8- to more than 3.5-fold increase in expression of several immunoglobulin heavy and light chains ($p < 0.01$), a 2.6- to 3.5-fold increase in expression of acute phase proteins (APPs) (alpha-2 macroglobulin (A2M), serotransferrin (TF/TRFE), ceruloplasmin (CP), clusterin (CLU), serum albumin (ALB), and transthyretin (TTR)) ($p < 0.01$), a 2.7- to 4.0-fold increase in expression of various glycoproteins, including alpha-1-acid glycoprotein (ORM1/A1AG1), alpha-1-acid glycoprotein 2 (ORM2/A1AG2), and alpha-1-B glycoprotein (A1BG) ($p < 0.04$), a 3.2-fold increase in expression of alpha-1-antichymotrypsin (A1ACT/SERPINA3) ($p < 0.01$), alpha-1-antitrypsin (A1AT/SERPINA1) and a 1.4- to seven-fold increase in expression of hemoglobin subunit alpha-1 (HBA1), hemoglobin subunit beta (HBB), and haptoglobin (HP/HPT) ($p < 0.01$). By way of contrast, the
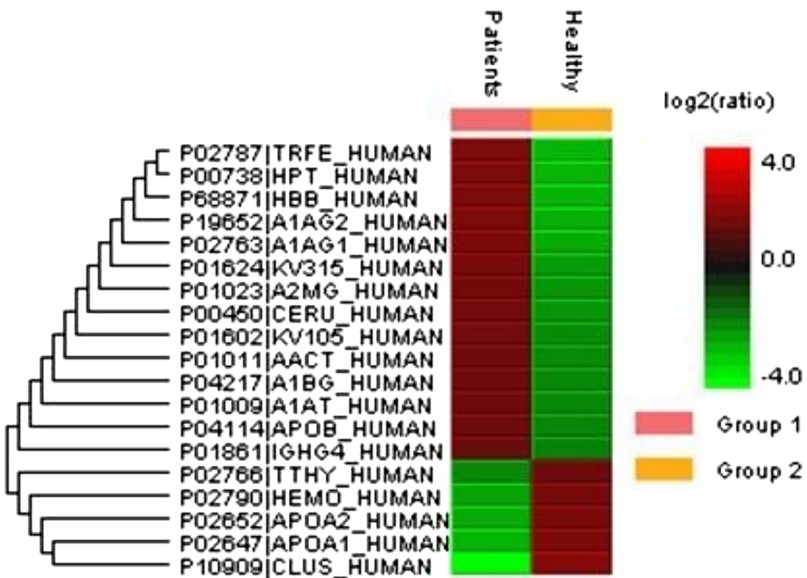

**Figure 2 An overview of the plasma protein profile of the SLE patients.** The heat map was generated using PEAKS Studio 8.0 displaying the differentially expressed proteins identified using LC-MS/MS label-free proteomic analysis between SLE patients and controls. The color scale representing the relative expression level of each protein across SLE and controls; red and green colors indicate the higher and lower levels of expressions. The intensity of the color represents the degree of protein up- and down-regulation when SLE patients and controls are compared.

expression of clusterin (CLU), apolipoprotein A-1 (APOA1), apolipoprotein A-2 (APOA2), and transthyretin (TTHY/TTR) were significantly decreased in SLE patients by 0.2- to 0.4-fold ($p < 0.01$).

## Functional annotation of the identified proteins

To gain valuable insight of the identified differentially expressed proteins from our analysis the international standardized gene function classification system of GO and the DAVID database were employed. When analyzed for CC analysis by GO annotations, most of the identified differentially expressed proteins were found in the extracellular space (46%), while others are macromolecular in nature (27%) and yet others were found to be part of membrane complexes (27%) (Fig. 3A). As shown in Fig. 3B, the BP evaluations of the GO annotations revealed that the identified proteins are largely involved in biological regulation (15.5%), complement activation (9.5%), negative regulation of the immune response (12.1%), multicellular organismal processes (14%), immune system processes (8.6%), metabolic processes (12.1%), the acute inflammatory response (10.3%), and localization (14.7%). On the other hand, when analyzed in terms of MF, the identified proteins were mostly involved in binding (41.2%), catalytic activity (38.2%), receptor activity (14.7%), transporter activity (14.7%), and serine peptidase activity (7.2%) (Fig. 3C; Tables S3 and S4). Therefore, based on the functional analysis, the identified proteins seem to primarily play important roles in immune system regulation ($p = 0.001$), in complement system activation ($p = 0.01$), in the innate and adaptive immune system responses ($p = 0.003$) and, potentially, in the acute inflammatory response ($p = 0.002$).

**Table 3 The list of statistically significant up and down-regulated proteins ($p < 0.05$, 0.01) between SLE patients and healthy control plasma samples.**

| Protein name | Gene name | Uniprot accession[a] | Mascot score[b] | Matched peptides[c] | MW[d] | Protein sequence coverage[e] | Average area of the triplicate analysis[f] | | Fold change SLE/controls | EmPAI (triplicates average) | |
|---|---|---|---|---|---|---|---|---|---|---|---|
| | | | | | | | SLE | Controls | | SLE | Controls |
| **Immunoglobulins** | | | | | | | | | | | |
| Ig kappa chain C variable region | IGKC | KV315_HUMAN | 212 | 77 | 12.49 | 42.7 | 81657.3 ± 30951.1 | 28738.8 ± 2422.5 | 2.84 | 0.90 ± 0.1 | 0.382 ± 0.01 |
| Ig heavy chain G4 | IGHG4 | IGHG4_HUMAN | 141 | 17 | 35.4 | 23.97 | 407207.9 ± 142838.3 | 118916.6 ± 10435.2 | 3.42 | 0.72 ± 0.1 | 0.29 ± 0.03 |
| Ig kappa variable chain 105 | IGKV1 | KV105_HUMAN | 132 | 11 | 12.8 | 28.9 | 86556.5 ± 32535.1 | 8556.5 ± 5003.5 | 10.1 | 4.51 ± 0.23 | 0.27 ± 0.02 |
| **Acute phase proteins** | | | | | | | | | | | |
| Serotransferrin | TF | TRFE_HUMAN | 2792 | 37 | 77 | 80.8 | 313293.8 ± 44564.8 | 134161 ± 24375 | 2.3 | 3.28 ± 0.1 | 1.19 ± 0.03 |
| Ceruloplasmin | CP | CERU_HUMAN | 1093 | 26 | 122.4 | 45.2 | 16466.94 ± 48797.0 | 45533.827 ± 6382.4 | 3.6 | 0.37 ± 0.03 | 0.16 ± 0.03 |
| Apolipoprotein B | APOB | APOB_HUMAN | 345 | 13 | 515 | 35.8 | 31774.2 ± 14099.0 | 8020.16 ± 931.8 | 4 | 0.26 ± 0.05 | 0.11 ± 0.01 |
| Clusterin | CLU | CLUS_HUMAN | 189 | 12 | 52.4 | 12 | 30586.1 ± 12888.3 | 8213311212.8 ± 0.3 | 0.4 | 1.15 ± 0.06 | 1.74 ± 0.21 |
| Apolipoprotein A1 | APOA1 | APOA1_HUMAN | 161 | 10 | 30.77 | 16 | 59486.2 ± 15316.8 | 122721.61 ± 17765.9 | 0.5 | 39.45 ± 6.10 | 142.53 ± 8.2 |
| Apolipoprotein A2 | APOA2 | APOA2_HUMAN | 56 | 2 | 11.75 | 53.33 | 41611.2 ± 13333.1 | 67031.6 ± 7289.8 | 0.6 | 43.21 ± 5.1 | 213.42 ± 9.12 |
| Alpha-2-macroglobulin | A2M | A2MG_HUMAN | 1290 | 77 | 163.2 | 47.8 | 6870155.6 ± 1014642.0 | 1812702.1 ± 203016.6 | 3.8 | 3.90 ± 0.25 | 1.33 ± 0.1 |
| Transthyretin | TTR | TTHY_HUMAN | 526 | 81 | 15.9 | 100 | 141897.8 ± 32858.1 | 206974.9 ± 13351.4 | 0.6 | 1.10 ± 0.4 | 6.22 ± 0.3 |
| **Glycoproteins** | | | | | | | | | | | |
| Alpha-1-acid glycoprotein 1 | ORM1 | A1AG_HUMAN | 210 | 79 | 23.51 | 23.9 | 181401.5 ± 70555.5 | 63420.2 ± 556.6 | 2.9 | 2.27 ± 0.44 | 0.950 ± 0.03 |
| Alpha-1-acid glycoprotein 2 | ORM2 | A1AG2_HUMAN | 140 | 75 | 23.60 | 13 | 145187.4 ± 48088.9 | 64026.6 ± 3758.3 | 2.3 | 1.95 ± 0.39 | 0.967 ± 0.03 |
| Alpha-1-B glycoprotein | A1BG | A1BG_HUMAN | 160 | 56 | 54.2 | 18 | 74140.823134 ± 0.7 | 31670.2 ± 4148.3 | 2.34 | 1.18 ± 0.1 | 0.47 ± 0.04 |
| **Antiproteases** | | | | | | | | | | | |
| Alpha-1-antitrypsin | SERPINA1 | A1AT_HUMAN | 96 | 35 | 46.7 | 30.1 | 265406.2 ± 49373.2 | 83843.4 ± 8059.6 | 3.2 | 21.41 ± 0.72 | 5.85 ± 0.17 |
| Alpha-1-antichymotrypsin | SERPINA3 | AACT_HUMAN | 82 | 11 | 47.6 | 30.1 | 38590.6 ± 17174.4 | 16122.3 ± 1641.8 | 2.9 | 0.71 ± 0.2 | 0.57 ± 0.3 |
| **Binding proteins** | | | | | | | | | | | |
| Hemopexin | HPX | HEMO_HUMAN | 72 | 7 | 51.67 | 22.9 | 29538.1 ± 8529.0 | 124043.9 ± 9040.8 | 0.2 | 1.0 ± 0.01 | 1.7 ± 0.1 |
| Hemoglobin beta subunit | HBB | HBB_HUMAN | 81 | 9 | 16 | 42.9 | 221930.6 ± 78425.9 | 83935.03 ± 16376.4 | 2.64 | 3.27 ± 0.2 | 0.90 ± 0.01 |
| Haptoglobin | HP | HPT_HUMAN | 176 | 26 | 45.2 | 30.5 | 276225.3 ± 89525 | 121576.9 ± 19207 | 2.27 | 5.10 ± 0.2 | 1.09 ± 0.01 |

| Protein name | EmPAI fold change | Protein regulation Up(+), Down (−) | Function | p-value[g] |
|---|---|---|---|---|
| **Immunoglobulins** | | | | |
| Ig kappa chain C variable region | 2.35 | + | Immune response and regulation | 0.04 |
| Ig heavy chain G4 | 2.47 | + | Immune response and regulation | 0.02 |
| Ig kappa variable chain 105 | 16.7 | + | Immune response and regulation | 0.01 |
| **Acute phase proteins** | | | | |
| Serotransferrin | 2.2 | + | Iron transport | 0.004 |
| Ceruloplasmin | 2.3 | + | Iron transport | 0.01 |
| Apolipoprotein B | 2.4 | + | Lipid metabolism | 0.04 |
| Clusterin | 0.6 | − | Innate immune response | 0.006 |
| Apolipoprotein A1 | 0.2 | − | Lipid transport | 0.009 |
| Apolipoprotein A2 | 0.2 | − | Lipid transport | 0.04 |
| Alpha-2-macroglobulin | 2.93 | + | Negative regulation of complement activation | 0.0003 |
| Transthyretin | 0.1 | − | Transporting thyroxine and retinol | 0.03 |
| **Glycoproteins** | | | | |
| Alpha-1-acid glycoprotein 1 | 2.3 | + | Acute phase, inflammatory response | 0.04 |
| Alpha-1-acid glycoprotein 2 | 2.03 | + | Acute phase response, inflammatory response | 0.04 |
| Alpha-1-B glycoprotein | 2.5 | + | Platelet degranulation | 0.03 |
| **Antiproteases** | | | | |
| Alpha-1-antitrypsin | 3.65 | + | Acute phase, inflammatory response | 0.02 |
| Alpha-1-antichymotrypsin | 1.24 | + | Acute phase, inflammatory response | 0.04 |
| **Binding proteins** | | | | |
| Hemopexin | 0.5 | − | Heme binding, transporter metabolism | 0.001 |
| Hemoglobin beta subunit | 3.63 | + | Heme binding, oxygen binding | 0.04 |
| Haptoglobin | 2.6 | + | Hemoglobin binding | 0.04 |

**Notes:**

The protein abundance differences among two groups were quantified using students $t$-test.

[a] Uniprot data entry.

[b] Mascot protein score revealed by MudPIT scoring. The integrated levels of expression. All the identified matches are significant with a significance level of 99% ($p < 0.01$) were considered a positive match when there are at least two unique peptides corresponding to the significance threshold with an ion score of 70.

[c] Number of matched peptides used to identify the protein. At least one matching peptide for each identified protein must fulfill the significance criteria ($p < 0.01$) and also be unique.

[d] Molecular weight in kDa.

[e] Number of peptides matched with a threshold significance value of $p < 0.05$.

[f] The matched peptide features (area under the curve) intensity of the patients compared to the controls.

[g] The statistical significance value after protein quantification data analysis.

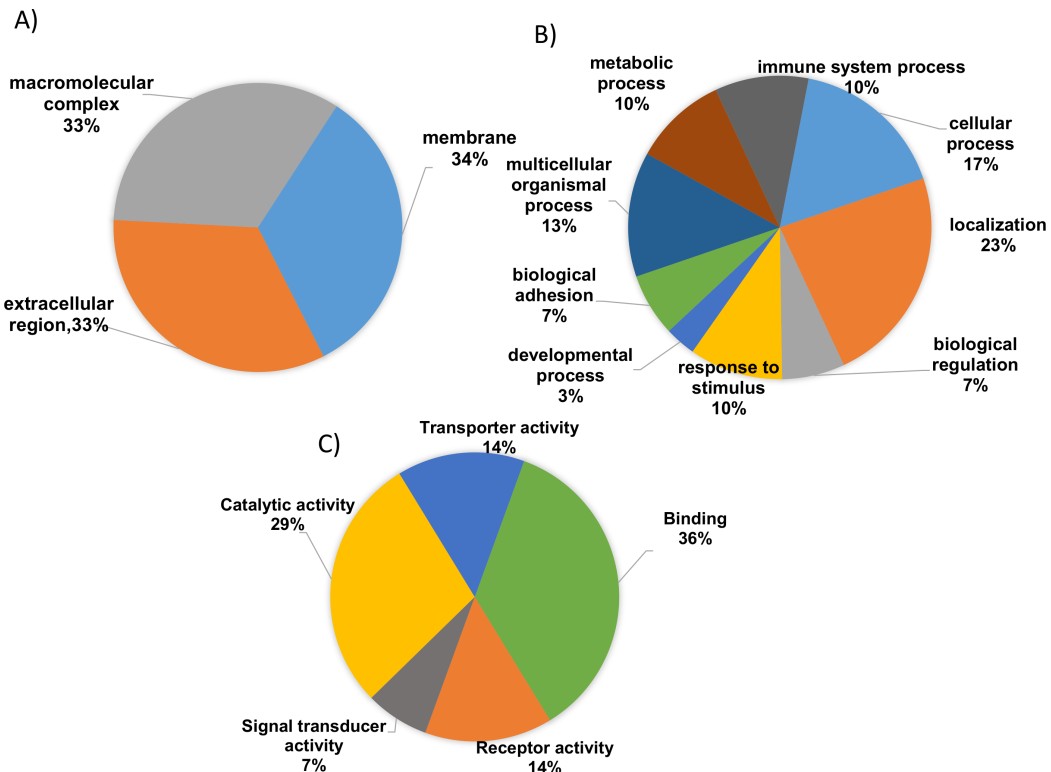

**Figure 3 Gene ontology (GO) enrichment analysis of the differentially expressed proteins.** (A) Cellular component analysis of the identified proteins. (B) Biological function. (C) Molecular function of the identified proteins. The pie charts were generated using Panther version 11.0 released 2016-07-15.

According to KEGG pathway annotation, the identified protein cohort was classified into more than 25 pathways (Tables S5 and S6) with the top 18 of them being shown in Fig. 4. Among 19 identified differentially expressed proteins, nine proteins (A2M, A1AT, A1ACT, CLU, KV105, KV315, IGG4, SERPING1, TF) were involved in complement and coagulation pathway, 12 proteins (A1AT, A1BG, HBB, TRFE, APOA1, AACT KV315, CLUS, KV105, A2MG, APOB, A1AG2) were involved in homeostasis and immune system processes ($p = 8.8 \times 10^{-8}$). In addition to the above findings, the majority of them were involved in a number of potentially interesting pathways such as scavenging heme from plasma, platelet activation, acute-phase responses, degranulation, signaling, aggregation, and the creation of C4 and C2 activators, etc. These pathways are highly relevant to the present study since SLE is an autoimmune disease caused by abnormalities that affect the immune system and in turn cause an acute inflammatory response. Therefore, the identified proteins are likely to play a significant role in SLE disease development and/or pathogenesis.

Additionally, the STRING PPI network analysis demonstrated a tight PPI network when the medium PPI confidence score of 0.4 was applied in STRING database, whereas at the high confidence score of 0.7 a significant association of PPT networks was identified among the differentially expressed proteins as shown in Fig. 5. This PPI network
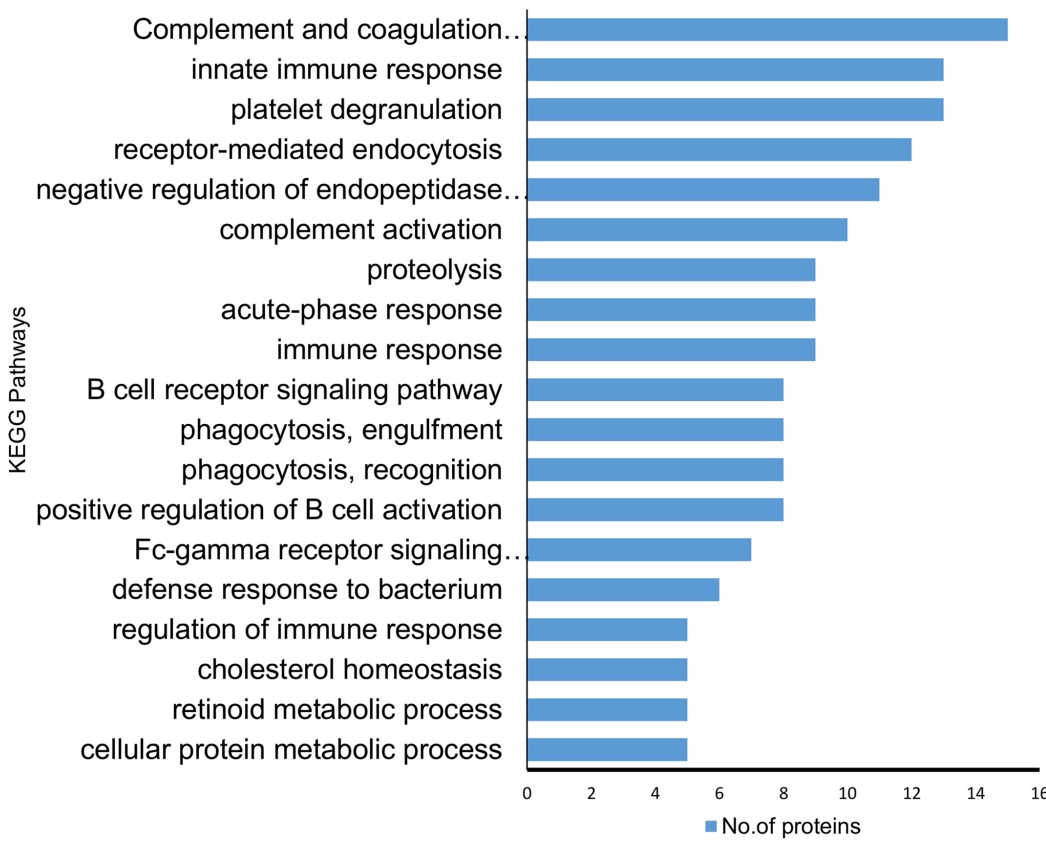

**Figure 4 Kyoto encyclopedia of genes and genomes (KEGG) pathway analysis of the proteins identified showing differential expression.** The majority of the identified proteins were enriched in relation to the complement and coagulation cascades and acute immune responses. The horizontal bars represent the number of differentially expressed proteins involved in various pathways.

suggests that the identified proteins have strong interactions with the proteins that contribute to numerous BP and have a wide range of MFs.

## DISCUSSION

Our comparative mass-based proteomic analysis investigated the proteomic profiles of SLE plasma samples were compared with healthy controls using the LC-ESI-MS/MS technique with label-free quantification. We were able to identify 122 differentially expressed proteins, and among these 19 proteins showed significant changes in expression with 14 of these being up-regulated and five being down-regulated in the SLE samples. When these proteins were subjected to the functional analysis, this protein profile was predominantly enriched with biological regulation, catalytic, and binding activity. Based on the KEGG pathway analysis most of the identified proteins particularly (A2M, A1AT, A1ACT, CLU, KV105, KV315, IGG4, SERPING1, and TF) significantly involved in complement coagulation cascades, regulation of innate and adaptive immune systems, inflammatory responses, platelet activation, and transportation of small molecules.

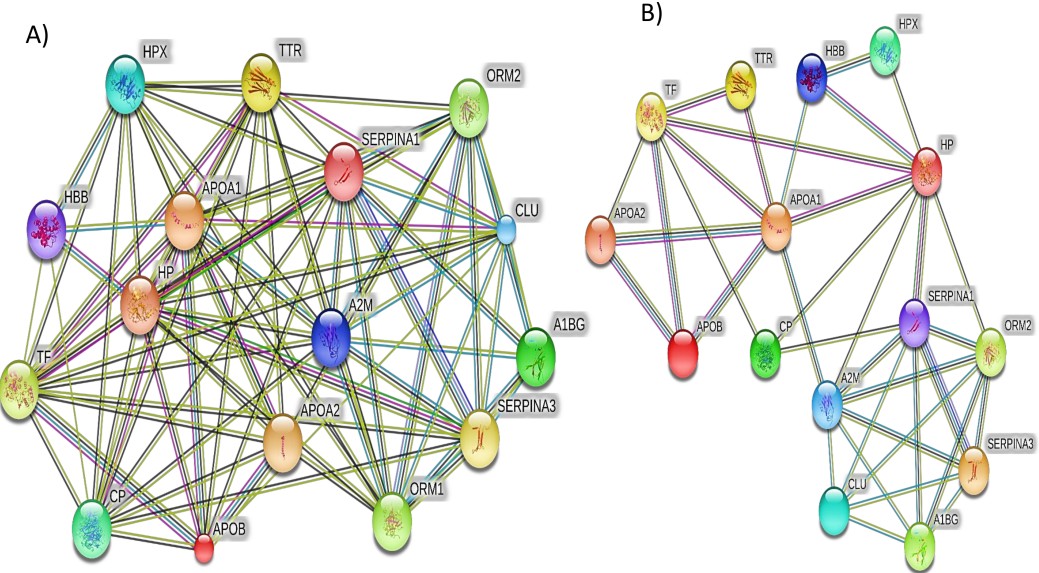

**Figure 5** **The protein–protein interactions for the differentially expressed proteins identified by LC-MS/MS label-free proteomics were analyzed using STRING software V9.1.** In the network analysis the differentially expressed proteins were represented as nodes. Each color of the lines connecting the nodes indicates strong evidence of the tight network of proteins. (A) The tight protein–protein interaction network obtained when the medium confidence level of 0.4 was applied. (B) The high confidence (0.7) PPI network of the identified significant proteins in SLE.

A set of nine APPs were up-regulated (A2M, A1AG, ORM1, A1AT, A1ACT, CP, APOB, HP TF) and two were down-regulated (APOA1, APOA2) in our analysis. These proteins were predominantly responsible for acute phase reactions including inflammation and dysregulation of the immune system (*Simon, 2013*). In addition to this, APPs also play active roles in various physiological and pathological activities related to disease exacerbation (*Simon, 2013*). It seems likely that the identified some of the identified APPs (A2M, A1AT, and A1ACT) play an important role in the complement and coagulation cascades which is an important pathway that plays a crucial role in SLE pathogenesis. Therefore, these proteins may be able to help our understanding of the mechanisms of immune reactions and inflammatory responses involved in SLE.

Alpha-2 macroglobulin protein has been previously found to be involved in a number of different diseases, including nephrotic syndrome, diabetes, and liver cirrhosis (*der Velden et al., 1998*). This is the first time we are reporting the higher abundance of A2M were identified in SLE patients compared to healthy individuals. In addition to these, several immunoglobulins heavy, and light chains were also found to be up-regulated in this study, and these changes illustrate that there is an imbalance in the immune system and homeostasis during SLE.

APOB have been shown to play a significant role in immune system regulation and inflammation. It was previously identified in the lupus patients with cardiovascular diseases and myocardial infarction, along with changes in low and high-density lipoproteins (LDL, HDL) (*Bots et al., 2016*; *Smržová et al., 2014*). Our study also identified

consistently up-regulated APOB expressions in SLE. We believe that this is the first report that shows increased expressions of APOB protein in SLE patients without any cardiac complications.

The AAP proteins which are down-regulated in this study are APOA1 and APOA2, these are major constituents of the high-density lipoprotein (HDL) complex, which has a specific role in lipid metabolism and has anti-inflammatory properties during both acute and chronic inflammation. Moreover, these are known to be immune regulators and suppress the generation of pro-inflammatory cytokines by activated T-cells (*Barry et al., 2004*) have been demonstrated in various autoimmune diseases. Furthermore, previous proteomic studies of RA and SLE have reported down-regulated APOA1 expressions may responsible for the active phase of the disease. These proteins also seem to have a key role in anti-inflammation (*Abe et al., 2001*; *Kazemipour et al., 2015*). Our results on APOA1 expression levels agree with the previous studies and seem to confirm the active SLE disease state of the patients used in this study.

A great deal of evidence has demonstrated that extreme oxidative stress leads to severe inflammation. Moreover, SLE is known to cause severe persistent inflammation in many major organs of the body. If we examine earlier SLE proteomic studies, some of the identified protein expressions by this study are consistent with previous reports, examples being ORM1/A1AG, TF, and CP, all of which have been detected in urine and renal biopsy samples obtained from SLE samples and were reported to act as biomarkers for class IV LN (*Brunner et al., 2012*; *Alaiya et al., 2015*; *Suzuki et al., 2007*). Our study identified the similar changes in expression of these proteins in our plasma SLE samples.

In addition to these, it has been shown that increased expression of SERPINA3/A1ACT and A1AT/SERPINA1 seems to play an important role in moderating inflammatory responses, and reducing the production of inflammatory mediators and the blockage of inflammatory cells. Aggarwal et al. recently reported the increased expressions of SERPINA3/A1ACT in LN patients and demonstrated A1ACT as the best marker to differentiate active renal lupus patients from active non-renal patients (*Aggarwal et al., 2017*). Thus, the identified differentially regulated antiprotease class of proteins may be relevant to renal dysregulation and inflammation in active lupus patients.

HP is also involved in immune system regulation, homeostasis, and tissue repair. It has been shown to be responsible for transporting hemoglobin, inhibiting iron loss and preventing kidney damage when hemolysis is occurring (*Galicia & Ceuppens, 2011*). Another study found higher expression of HP in SLE patients may contribute to hypergammaglobulinemia, systemic vasculitis and cardiovascular disorders (*Pavón et al., 2006*). Therefore, the observed HP changes from this study is consistent with the findings of the previous study and suggests that the increased HP expressions may cause severe clinical manifestations in SLE.

Two hemoglobin scavenger proteins such as HBB and HPX showed differential regulations from our findings were implicated with HDL and influence the inflammatory properties of HDL and scavenging of oxygen binding and transport from the lungs to the peripheral tissues and also involved in a number of inflammatory diseases

(*Newkirk et al., 1999*). In addition, the depletion of HPX levels implicated in a number of inflammatory diseases such as septic shock and experimental autoimmune encephalomyelitis (*Mehta & Reddy, 2015*). The primary function of HPX is heme scavenging and it protects against oxidative stress and related inflammatory diseases. Moreover, the precise role of these proteins in SLE is not recognized yet, and to the best of our knowledge this is first plasma proteomic study reporting the down-regulated expressions of HPX and up-regulated HBB in lupus patients.

The two more down-regulated proteins from our analysis are CLU, TTR/TTHY which are also potentially associated with SLE pathogenesis. These play crucial roles in immune system dysregulation and are associated with alterations in inflammatory reactions. CLU is a glycoprotein with ubiquitous tissue distribution that has been reported to be implicated in several physiological processes. Earlier evidence has shown that lower levels of CLU/apoJ are associated with some feature of SLE such as a diminished control of antibody-mediated inflammation at the sites of apoptosis where auto-antigens are exposed (*Andrade, Casciola-Rosen & Rosen, 2000*). Moreover, CLU may have a protective role against SLE disease activity because disturbances in apoptosis and complement function seem to play essential roles in SLE pathogenesis (*Burger & Dayer, 2002*). The down-regulated expression level of CLU in SLE patients identified by our analysis might be associated with the fact that the patients studied had active disease. Therefore, the above findings on CLU expression levels among SLE patients suggest that this protein is likely to be an important biomarker with respect to SLE disease activity.

Another down-regulated protein identified is TTR, which is a serum and cerebrospinal fluid transporter of the thyroid hormone thyroxine (T4), and of retinol (*Zheng et al., 2001*). Earlier proteomic observations by Rana et al. showed the increased expression of TTR in serum obtained from pediatric SLE patients compared to controls (*Rana et al., 2012*). In our proteomic study, we have identified TTR reduced expressions in adult SLE patients.

## CONCLUSION

Taking the above as a whole, we have demonstrated that the identification of a single protein that possesses exclusive characteristics of an SLE marker is indeed very challenging. The identification of differential expressions of proteins in the plasma between SLE patients and controls will help to develop novel approaches for the early disease detection, prevention, and the treatment of SLE. Our findings have revealed that a number of proteins show changes in expression associated with the presence of SLE. Moreover, some of the identified proteins and their expressions from this study support the findings of previous studies. Additionally, our findings may play a significant role in inflammation, acute phase responses, disease activity and immune dysregulation when SLE is present. Nevertheless, these interesting findings would require subsequent investigation to validate and confirm. Finally, LC-MS/MS combination with label-free quantification is clearly an excellent technique for the profiling of differentially regulated proteins in the disease samples.

## ABBREVIATIONS

| | |
|---|---|
| **SLE** | Systemic lupus erythematosus |
| **LN** | Lupus nephritis |
| **LC-ESI-MS/MS** | Liquid chromatography electrospray ionization tandem mass spectrometry |
| **EmPAI** | Exponentially modified protein abundance index |
| **BPC** | Base peak chromatogram |
| **XICs** | Extracted ion chromatograms |
| **AUC** | Area under the curve |
| **A2M** | Alpha 2 macroglobulin |
| **TTR/TTHY** | Transthyretin |
| **HPT/HP** | Haptoglobin |
| **APOB** | Apolipoprotein B-100 |
| **GO** | Gene ontology |
| **KEGG** | Kyoto encyclopedia of genes and genomics |
| **DAVID** | Database Annotation Visualization, and Integrated Discovery |
| **STRING** | Search Tool for the Retrieval of Interacting Genes/Proteins, Version 9.1 |
| **PPI** | Protein–protein interaction networks |
| **APPs** | Acute phase proteins |
| **CC** | Cellular components |
| **MF** | Molecular function |
| **BP** | Biological process |

## ACKNOWLEDGEMENTS

We would like to thank Cathay General Hospital (CGH) for providing the SLE and healthy plasma samples for our proteomic analysis.

### Funding

This work was supported by grants from the National Central University and CGH collaborative project. The grant registration numbers are 104G909-4, 10310061-4, 10210061-3, VGHUST 105-G1-3-2, VGHUST 106-G1-3-2 and MOST 106-2320-B-008-004-MY3. The funders had no role in study design, data collection and analysis, decision to publish, or preparation of the manuscript.

### Grant Disclosures

The following grant information was disclosed by the authors:
National Central University and CGH collaborative project: 104G909-4, 10310061-4, 10210061-3, VGHUST 105-G1-3-2, VGHUST 106-G1-3-2, MOST 106-2320-B-008-004-MY3.
## Competing Interests

The authors declare that they have no competing interests.

## Author Contributions

- Rashmi Madda conceived and designed the experiments, performed the experiments, analyzed the data, prepared figures and/or tables, approved the final draft.
- Shih-Chang Lin authored or reviewed drafts of the paper, approved the final draft, samples provider.
- Wei-Hsin Sun contributed reagents/materials/analysis tools, authored or reviewed drafts of the paper, approved the final draft.
- Shir-Ly Huang conceived and designed the experiments, analyzed the data, contributed reagents/materials/analysis tools, authored or reviewed drafts of the paper, approved the final draft.

## Human Ethics

The following information was supplied relating to ethical approvals (i.e., approving body and any reference numbers):

This study was approved by the Institutional Review Board (IRB) for Research Ethics at Cathay General Hospital, Taiwan (IRB project identification code: CT-099005).

## Data Availability

Madda, Rashmi; Lin, Shih-Chang; Sun, Wei-Hsin; Huang, Shir-Ly (2018): Plasma proteomic analysis of systemic lupus erythematosus patients using liquid chromatography/tandem mass spectrometry with label-free quantification. figshare. Fileset. https://doi.org/10.6084/m9.figshare.5970748.v1.

## Supplemental Information

Supplemental information for this article can be found online at http://dx.doi.org/10.7717/peerj.4730#supplemental-information.

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
