# Peer review of "Plasma proteomic analysis of systemic lupus erythematosus patients using liquid chromatography/tandem mass spectrometry with label-free quantification"

_PeerJ, doi:10.7717/peerj.4730_

## Round 0.1 · original submission · Major Revisions

Please pay special attention to the critical comments made by reviewer 1. Make sure the English language is well revised and ensure the statistical remarks are answered and incorporated in a proper way.

Reviewer 1 ·

Basic reporting

1. The English language should be improved and hence an extensive revision of English by native speaker is recommended (e.g. lines 355, 357-359, 368-371, 378-379, 395-397).

2. The discussion needs to be reorganized and more concise. The acute phase proteins should be discussed together (combine line 355-367 and 391-399 with line 327-334). Next, another proteins connected to inflammation should be mentioned and at the end of the
discussion others proteins should be discussed. It is good to group discussed proteins in relation to their similar function/pathway/process.

3. The mass spectrometry data should be available online and deposited to the ProteomeXchange Consortium via the PRIDE partner repository.

4. The information about standard deviation (SD) should be added to Table 3 as it is stated in method section (data are presented as a mean + SD).

5. Labels A, B, C should be added to Figure 4 and 6.

6. KEGG pathway analysis (Figure 5) indicated that 9 differentially expressed proteins were connected to complement activation. However, there is no information in the discussion which of identified proteins (and why) are engaged in complement activation. This could be potentially important in the pathogenesis of SLE. The same issue is for proteins related to coagulation.

Experimental design

No comment

Validity of the findings

1. My main concerns regarding this paper are about statistical analysis of the data. In method section the authors did not mention that they performed p-value adjustment with Benjamini-Hochberg False Discovery Rate (FDR) (p<0.05) correction in order to test for multiple hypothesis. If it is not tested, the obtained results could not be reliable.

2. Why did the authors use three statistical test (independent t-test, paired-t-test, ANOVA) to determine the differentially expressed proteins between two groups? And p-values presented in Table 3 are related to which test?

3. How could the authors explain the huge discrepancies in fold changes between presented two methods of analysis for clusterin and hemoglobin beta subunit?

Reviewer 2 ·

Basic reporting

1. Language is generally clear and professional. A couple of places need attention for language: (1) line 240 first paragraph of the Results section, there seems to be a word missing "the proteins at least two of the three..." (2) line 325, first paragraph of the discussion also seems to have a word missing in phrase "significantly enriched by involved in the innate..."
2. Literature references are sufficient. However reference #17 is an abstract, not a published article, and I would suggest that it is not suitable to cite unless there is a related full publication.
3. Structure, figures and tables are OK with the exception of Figure 1: for this type of study it is not necessary to show such a diagram and it can be eliminated.
4. Data are shared.
5. Relevant results presented; hypotheses presented.

Experimental design

1. This is original primary research.
2. The research question is well-defined and it does fill a gap though it is in large part just confirmatory of other studies.
3. Technical approaches are appropriate; use of human subjects is ethical.
4. Methods are describe in sufficient detail.
5. Need to get more information about antibodies to dsDNA; the text states that all the patients were positive but the table of this information (Table 1) indicates a range of values. Furthermore, we need to know what is considered the normal range for the laboratory.
6. In table 1 it is not necessary to indicate "disease status" as the SLEDAI score, which is elevated, confirms the active disease condition.

Validity of the findings

1. Findings appear valid; as the authors note these findings will need replication by another approach or in another patient group to confirm.
2. Data are appropriately controlled; the patients are compared to a control group with similar features.
3. Conclusions are clear.
4. The utility of these findings as prognostic/diagnostic markers is at this stage speculative but possibly this is a direction for future research and the conclusion in this regard, as stated in the abstract, is appropriate.

Additional comments

In general these findings demonstrate utility of a technique that might uncover novel or informative markers and pathways in lupus. At the present stage, the results are not especially novel; the pathways uncovered are ones that are generally well-recognized to contribute to lupus pathogenesis, such as those associated with complement and the acute phase response. It is not clear yet that such an approach has diagnostic or prognostic utility over existing and available tests.

---

## Round 0.2 · accepted · Accept

The reviewer was happy with the thorough revisions you have included (see below) and hence we are now happy to accept your paper for publication. I would like to congratulate you!

# Reviewer 1 ·

Basic reporting

The authors revised the paper thoroughly according to my suggestions and I do not have additional comments.

Experimental design

No comment

Validity of the findings

The authors revised the paper thoroughly according to my suggestions and I do not have additional comments.